# Missed Nursing Care; Prioritizing the Patient’s Needs: An Umbrella Review

**DOI:** 10.3390/healthcare12020224

**Published:** 2024-01-16

**Authors:** Iokasti Papathanasiou, Vasileios Tzenetidis, Konstantinos Tsaras, Sofia Zyga, Maria Malliarou

**Affiliations:** 1Department of Nursing, University of Thessaly, 41500 Larisa, Greece; iokpapathan@uth.gr (I.P.); vtzenetidis@uth.gr (V.T.); ktsa@uth.gr (K.T.); 2Department of Nursing, University of the Peloponnese, 22131 Tripolis, Greece; zygas@uop.gr

**Keywords:** missed care, prioritization, nurses

## Abstract

The objective of this review of reviews was to identify the reasons for missed nursing care and to shed light on how nurses prioritize what care they miss. Missed nursing care refers to essential nursing activities or tasks that are omitted or not completed as planned during a patient’s care. This omission can result from various factors, such as staffing shortages, time constraints, or communication issues, and it can potentially compromise the quality of patient care and safety. Identifying and addressing missed nursing care is crucial to ensure optimal patient outcomes and the well-being of healthcare professionals. To be included, reviews had to use the systematic review process, be available in the English language, examine missed care in hospitals and at home, and include participants who were over eighteen years old. The review intended to answer the following questions: ‘Why nursing care is missed?’ ‘How nurses prioritize what care they missed?’. An umbrella review was developed guided by the JBI methodology and using PRISMA-ScR. A total of 995 reviews were identified. According to the inclusion criteria, only nine reviews were finally evaluated. The findings indicate that care is missed due to staffing levels, organizational problems, and the working climate. Prioritization of care depends on acute care needs as well as educational and experiential background. Missed nursing care is associated with patient safety and the quality of provided nursing care. Specifically, it has negative impacts on patients, healthcare professionals, and healthcare service units. Organizational characteristics, nursing unit features, and the level of teamwork among nursing staff affect Missed Nursing Care. Individual demographic characteristics of the staff, professional roles, work schedules, and adequate staffing may potentially contribute to the occurrence of Missed Nursing Care, which is why they are under investigation. However, further consideration is needed regarding the management of patient needs and nurse prioritization.

## 1. Introduction

The MNC (Missed Nursing Care) was first recognized in 2006 by Kalisch, who defined it as “when any part of the required care is omitted (in part or in whole) or delayed” [1]. It is an error due to omission. The error can be an act that has been omitted, such as not mobilizing the patient or an act that has been performed incorrectly, like marking the wrong eye for surgery [2]. It is a concept with three dimensions: first, the problem of resource and time scarcity; second, the decision-making process to prioritize and allocate nursing care; and third, the care that remains incomplete [3].

In the first qualitative research conducted by Kalisch, the activities that were found to be more frequently omitted include mobilization, changing bed positions, feeding, education, discharge planning, emotional support, hygiene, documentation of admissions and discharges, and monitoring [2]. Some of the reasons for care omissions are insufficient staff numbers [4,5,6], a high volume of patient admissions and discharges in the department, and an insufficient number of support or administrative personnel [7].

In the international literature, various terms have been used to describe this phenomenon, and there is no absolute agreement on what MNC means and how it is perceived by nursing staff [3]. The phenomenon is described as the non-provision or failure to complete necessary nursing tasks [8], unmet care needs [9], care that is not provided or not completed [10], a prioritization of care when resources are limited [11], and prioritizing the care provided [12].

The Kalisch Model of Missed Care analyzes the fundamental characteristics contributing to nursing care omissions as reported by nursing personnel [2]. In the healthcare environment, prior instances of missed nursing care conditions prompt nursing personnel to carefully consider the care they provide. Key factors influencing this decision-making include assessing the available human resources and considering the number, education level, and experience of nursing staff, along with support staff availability. Additionally, the presence of material resources, such as medications and functional equipment, significantly impacts staff effectiveness. Effective teamwork and communication are vital, both within the nursing unit and between medical and nursing staff, as well as with support staff. In situations where these conditions are compromised, nursing personnel engage in reassessment, prioritizing care based on available resources.

Facing such challenges, nursing personnel integrate their actions into the nursing process, which is characterized as a client-centered approach emphasizing organization, critical thinking, knowledge application, and decision-making. The decision to delay or omit a nursing action is influenced by internal factors, including group standards, which encompass informal rules and accepted behaviors within the team. Nursing staff also engage in priority-based decision-making, considering patients’ needs, conditions, health statuses, and other requirements. Values, perceptions, and attitudes of nursing staff regarding their roles and responsibilities play a significant role in determining which nursing actions take precedence. Additionally, habits formed over time may lead nurses to make decisions unconsciously. Comprehensively considering these internal and external factors, nursing personnel navigate the complex landscape of providing quality care within resource constraints [2].

MNC is related to patient safety and the quality of nursing care provided [3]. Specifically, it has negative impacts on patients, healthcare professionals, and healthcare service units. Concerning patients, it reduces safety [10] and the overall quality of care they receive [13]. It has been found that MNC is associated with decreased patient satisfaction with the care they receive [8,14,15], an increase in medication administration errors, increased hospital-acquired infections, pneumonia, falls, pressure ulcers, emergent situations [8,14,16,17], and in-hospital mortality in patients undergoing common surgical procedures [4]. Additionally, according to Schubert and colleagues, hospitalized patients in units with a high level of MNC have a 51% greater chance of mortality [18].

Regarding healthcare professionals, MNC is associated with decreased job satisfaction [19], role conflict, a sense of ethical burden [20], disappointment, anxiety, and dissatisfaction because the nursing staff is unable to practice their profession in alignment with their personal and professional values [21], and the provision of low-quality care [3].

At the level of healthcare service units, MNC is related to increased job turnover and higher rates of absenteeism [22].

According to the later model by Kalisch and Lee (2010), organizational characteristics, characteristics of the nursing unit, and the level of teamwork among nursing staff predict missed care (MNC). Individual demographic characteristics, professional roles, work schedules, and adequate staffing may potentially contribute to MNC and are therefore investigated [23]. Through teamwork in the workplace, safety is achieved, effective patient-centered healthcare delivery [24,25] is promoted, and errors are prevented [26]. Additionally, when exploring the relationship between nursing staff teamwork and MNC, international research findings indicate a significant correlation, specifically that effective teamwork reduces MNC [26,27].

The primary aim of conducting this review of reviews was to meticulously uncover the underlying reasons behind the phenomenon of missed nursing care and to illuminate the intricate processes that guide nurses in prioritizing the care they unintentionally omit.

## 2. Materials and Methods

### 2.1. Study Design

The overview of reviews showed a lot of studies on the subject, looking at it from different perspectives. For this reason, we started this umbrella review, following the preferred reporting items for scoping reviews (PRISMA-ScR), and were guided by the methodology proposed by JBI to adequately conduct umbrella reviews [28,29,30].

### 2.2. Eligibility Criteria

In the initial phase, we established inclusion and exclusion criteria for the review, which can be found in Table 1. Specifically, the inclusion criteria encompassed papers that addressed the following questions: Why nursing care is missed? How do nurses prioritize the care they miss?

Furthermore, the inclusion criteria for articles required adherence to specific parameters. These articles had to qualify as systematic reviews, be accessible in free full text, be composed in English, and be published within the timeframe of 2013 to 2023.

Conversely, exclusion criteria were distinctly outlined. Articles that failed to address the research inquiries involved healthcare professionals other than nurses (e.g., doctors, dentists), contained paid content, or fell into other review categories were excluded. Additionally, articles outside the publication period of 2013 to 2023 and those not freely accessible in the English language were also excluded from consideration.

It is crucial to mention that the escalating interest in missed nursing care is evident through a notable surge in publications over the past 15 years, encompassing a diverse array of review types since 2013. In addition, articles requiring payment were excluded, as the research was conducted without external funding and faced limitations in available resources. It is noteworthy that the decision to include only English-language texts was determined by the languages spoken and written by all participants involved in the study.

### 2.3. Data Collection

The second phase involved searching for and selecting all relevant records that met the criteria for the review. As outlined in Table 2, records were sourced from Scopus, Medline databases, and Cochrane. These databases were chosen for their relevance to nursing care reviews. To address the research questions, a set of common keywords was applied across all databases. Specifically, these keywords included ‘missed nursing care’, ‘unmet nursing care’, ‘unfinished care’, ‘rationed nursing care’, ‘prioritized nursing care’, ‘omissions in nursing care’, and ‘nursing care left undone’.

### 2.4. Data Processing and Analysis

Three researchers participated in the database search. The process was conducted independently by each researcher to ensure the integrity of the review. Initially, two of them focused on reading titles and abstracts, while the third served as a consultant and intervened when consensus could not be reached. All selected documents were thoroughly read, and an Excel document was created, including the following details: author, country, review type, databases used, number of studies, aim, quality appraisal, and sample size. This information was crucial for selecting and analyzing the content of the documents.

## 3. Results

From the research, 955 reviews were identified in the databases, as shown in Figure 1. In the first phase, duplicate studies and documents that did not meet the inclusion criteria were removed, resulting in the exclusion of 578 articles. In the second phase, after careful reading, 328 articles were further excluded as they did not meet the inclusion criteria. In the third phase, 49 remaining documents were read, of which 40 were excluded based on the inclusion criteria. Finally, nine reviews met the criteria.

Table 3 presents the characteristics of the reviews. The authors hailed from a diverse range of countries, including two from the UK, two from Sweden, one from Greece, one from Canada, two from Finland, and one from Germany. Five reviews utilized recognized quality appraisal tools.

Furthermore, eight reviews addressed the question of why care is missed, while four delved into how nurses prioritize the care they miss.

### 3.1. Why Nursing Care Is Missed

Table 4 summarizes the findings of nine reviews, showing the reasons why nursing care is missed. More specifically, studies have underscored the critical link between staffing levels and the quality of nursing care. One investigation revealed a significant correlation between lower nurse staffing levels and elevated instances of missed nursing care. Also emphasized this connection by highlighting that the hours nurses dedicated per patient daily were inversely associated with the occurrence of missed care. These findings underscore the pivotal role that adequate staffing plays in ensuring complete and timely nursing interventions [40]. The organizational aspects explored unveiled a noteworthy insight: a leading factor contributing to missed nursing care was the insufficient presence of staff or their improper deployment. This sheds light on the critical importance of effective staffing strategies in mitigating the occurrence of overlooked nursing responsibilities [31]. According to Imam A et al. [33], the primary contributing factor to the challenge was identified as an insufficient number of nursing staff, which ranked at the forefront. Following closely, inadequate availability of assistive personnel and an unforeseen surge in patient volume and/or acuity were both notable, securing the second position in the identified factors. These findings underscore the multifaceted nature of the issue, emphasizing the critical role of staffing adequacy in addressing challenges within the healthcare environment.

In certain instances, facing racism acted as a barrier, hindering both patients and their families from accessing essential treatment and care. The impact of discriminatory experiences manifested as an impediment to the delivery of necessary healthcare services [33]. When registered nurses focused on addressing patients’ acute care needs, the available time to attend to basic care needs became limited. Additionally, there were instances in the emergency department where essential care for existing patients was deprioritized as nurses awaited potential incoming patients with acute care needs. These examples underscore the complex balancing act that healthcare professionals navigate in managing competing priorities within their workflow [36]. The majority of the included studies detailed the impact of inadequate staffing and dimensioning of emergency departments (EDs) in relation to patient load, leading to instances of missed nursing care (MNC). Issues such as crowding, spatial limitations, and imbalances in staff-to-patient ratios were specifically highlighted as notable challenges associated with these circumstances [36]. Patients identified several staff-related factors contributing to perceived instances of missed care, encompassing a shortage of staff, insufficient staff experience, inadequate teamwork, a lack of communication among staff during shift changes, and the demeanor of staff members. These findings highlight the multifaceted nature of patient perceptions regarding missed care, pointing towards crucial aspects such as staffing levels, experience, teamwork, and communication that significantly influence the overall care experience [34]. The frequency of missed care activities demonstrated a direct correlation with the level of dissatisfaction among healthcare professionals. Moreover, a positive association was observed between burnout and instances of missed care. Additionally, a higher incidence of missed care activities was linked to an increased intention to leave the job, highlighting the intricate interplay between job satisfaction, burnout, and the quality of care provided [37]. Most studies highlighted the prioritization of nursing actions as one of the key factors in nursing care neglect [31,34,36,38].

### 3.2. How Nurses Prioritize What Care They Miss

The organization’s structure plays a pivotal role in guiding nurses when they prioritize tasks. Equally important is the nurses’ decision-making ability, influencing the determination of which care to administer and what might be omitted. Drawing on their education and experience, healthcare professionals navigate a delicate balance between the care needs of patients and the available resources. The prioritization of patients’ acute care needs takes precedence in this delicate equilibrium. However, in the face of limited resources, care that is deemed of lesser value to patients may find itself deprioritized, leading to instances where such care is ultimately missed or overlooked. This underscores the challenging decisions healthcare professionals must make to ensure optimal and essential care delivery within the constraints of available resources [36,39]. Moreover, it is influenced by the educational background of nurses and the practical knowledge they have gained through their work experiences [34]. According to Suhonen et al. [39], prioritization is driven by a commitment to address the diverse needs of their patients comprehensively and holistically; this approach manifests in various contexts. These include considerations such as patient groups, specific diseases, the severity of the patient’s condition, age, and the perceived benefits of the treatment [38] (Table 5).

## 4. Discussion

This comprehensive examination of nine systematic reviews offers an overview of the factors influencing why nurses miss care and the criteria guiding their decisions on which care to prioritize. Despite variations in labeling, such as systematic review and scoping review, all the included reviews adhered to the systematic review process, with many explicitly mentioning the application of the PRISMA guidelines.

The phenomenon of missed nursing care has been studied to a considerable extent in recent years. However, few approaches examine the issue comprehensively, focusing on the causes. A comprehensive assessment requires a deep understanding and the ability to evaluate and identify all factors of the problem. The majority of the studies used in the research refer to the reasons why nursing care is omitted, with only four addressing how nurses prioritize patient needs and based on the primary reasons they make this decision.

Summarizing the results regarding the causes of missed care, we observe that they are attributed to both issues within the organization itself and problems related to the individual nurse. Organizations must initially prioritize the protection of both patients and nurses. This can be achieved by ensuring all available resources for nursing care are provided. However, resources require sufficient and specialized personnel with knowledge and training capable of addressing challenging patient situations.

Consequently, it is imperative that the healthcare personnel not only meet the baseline requirement but also operate at their maximum potential, with a clear objective of prioritizing patient care. This involves ensuring that the staff is not only adequately sized but also effectively deployed to address the diverse needs of the patients.

Furthermore, an essential aspect of sustaining high-quality care is the continuous training of the staff. This training should encompass both clinical nursing skills and administrative competencies, ensuring that the healthcare team remains well-equipped to handle evolving medical practices and organizational demands. Regular training sessions contribute to the ongoing professional development of the staff, enhancing their ability to deliver optimal healthcare services.

In addition to the technical aspects, fostering a collaborative environment within the healthcare team is crucial. Maintaining a team spirit encourages open communication, knowledge-sharing, and a collective commitment to patient well-being. This collaborative culture goes beyond individual responsibilities, creating a cohesive unit where each member contributes to the overall efficiency of healthcare delivery.

Moreover, effective communication plays a pivotal role in providing patient-centered care. Clear and open lines of communication between healthcare professionals, as well as with the patients, contribute to better understanding and coordination. This, in turn, cultivates a sense of empathy and responsibility towards the patients among the healthcare staff. By promoting an environment where empathy is prioritized, healthcare providers are better able to connect with patients on a personal level, addressing not only their medical needs but also their emotional well-being.

In essence, ensuring the adequacy, optimal utilization, and continuous development of healthcare personnel, coupled with collaborative team culture and effective communication, are integral components in fostering a healthcare environment that prioritizes patient-centric care and addresses the holistic needs of individuals under their care.

Delving into the intricacies of nursing care prioritization reveals a nuanced landscape where factors such as the educational background of healthcare professionals and the severity of patients’ health conditions play pivotal roles. This exploration unveils that these elements are not merely incidental but stand out as fundamental determinants influencing whether a particular patient’s needs receive the requisite attention or are inadvertently neglected.

In essence, the educational attainment of healthcare practitioners emerges as a critical variable. The depth and breadth of their education significantly impact their ability to discern and prioritize the diverse needs of patients. A well-educated healthcare workforce is more likely to navigate the complexities of patient care with acumen, ensuring that critical needs are identified and addressed promptly.

Simultaneously, the gravity of patients’ health conditions serves as another cornerstone in the hierarchy of nursing care priorities. The severity of an individual’s health status inherently dictates the urgency and intensity of care required. In instances where patients face more acute or complex health challenges, there is a heightened need for vigilant and immediate attention to address their specific needs.

Moreover, it is imperative to recognize that these factors do not operate in isolation but often intersect and interact in dynamic ways. For instance, a healthcare professional’s level of education may directly influence their ability to gauge the severity of a patient’s condition accurately. Conversely, the severity of a health condition may, in turn, impact the healthcare provider’s decision-making process regarding the allocation of resources and attention.

In summary, the prioritization of nursing care is a multifaceted process, influenced significantly by the educational background of healthcare professionals and the severity of patients’ health conditions. Understanding and navigating this intricate interplay is crucial for fostering a healthcare environment where the diverse needs of patients are not only recognized but also addressed with precision and empathy.

Addressing the initial aspect of the review, it is evident that nurses indeed miss care, and we classify the reasons into four key domains: organizational factors, nursing-related factors, and patient and psychosocial conditions. Examining the organizational factors in-depth, we identified factors contributing to missed care, such as understaffing, inefficient staff deployment, a lack of teamwork, and communication gaps between shifts. When delving into nursing-related considerations, the focal points encompass staffing adequacy, the depth of professional experience, and the allocation of time dedicated to individual patients. These factors play pivotal roles in shaping the landscape of care delivery. In the context of patient considerations, it becomes evident that as the severity of a patient’s condition increases, the available time for attending to their fundamental needs diminishes. Within the psychosocial behaviors category, instances of missed care are influenced by nurses expressing racist motives towards patients with distinctive characteristics. Moreover, factors such as non-professional satisfaction, fostering the intention to leave the job, and professional disempowerment stand out as contributors to missed care.

Delving into the prioritization methods adopted by nurses in carrying out their responsibilities, we identified two distinct categories: factors related to nursing and those related to patients. More precisely, within the nursing category, we pinpointed critical factors shaping prioritization, including educational background, accumulated experiences, and the commitment to fulfilling all aspects of patient needs. As for the ‘patients’ category, it encapsulates considerations such as the severity of illness, age, and the moral satisfaction nurses derive from enhancing the health of their patients (Figure 2).

## 5. Conclusions

This comprehensive review meticulously navigates the intricate landscape of missed nursing care, unraveling a complex web of factors that shape its occurrence and shedding light on the decision-making processes employed by nurses. However, a deeper, more nuanced critical analysis is warranted to fully grasp the study’s breadth and implications.

To initiate this analysis, a thorough examination of the review’s limitations is imperative. While the insights provided are invaluable, acknowledging inherent constraints, including potential biases, scope restrictions, and methodological limitations, enhances the transparency and robustness of the findings. Addressing these limitations establishes a more realistic context for interpreting the results.

Furthermore, delving into the practical applications of the study’s findings is crucial. Understanding how the identified factors contributing to missed care align with real-world healthcare scenarios empowers practitioners to develop targeted interventions. By extrapolating implications for nursing practices, the study becomes a foundational resource for healthcare professionals striving to elevate patient care standards.

Additionally, a more expansive discussion on the practical implications of the prioritization methods identified in the review would be beneficial. Examining how these methods translate into actionable strategies within healthcare settings provides practical insights for nurses and healthcare administrators. Concrete examples from scientific literature and empirical evidence can illustrate both successful interventions and challenges encountered in implementing prioritization strategies.

Moreover, a nuanced exploration of the intersectionality of factors contributing to missed care is essential. Understanding how organizational factors, nursing-related considerations, patient conditions, and psychosocial behaviors intersect offers a holistic view. This analytical approach deepens comprehension of the complexities involved in addressing missed care within healthcare systems and contributes significantly to the ongoing discourse on enhancing nursing care practices and improving patient outcomes.

To provide a general interpretation of the results concerning the review questions and objectives, as well as potential implications, it is crucial to synthesize the key findings. This involves summarizing how the identified factors influencing missed nursing care align with the initial research questions and objectives. Additionally, exploring the broader implications of these findings for healthcare practices, policy-making, and patient outcomes will contribute to a comprehensive interpretation. This synthesis will provide a clearer understanding of the study’s significance and guide future directions in nursing research and practice.

## Figures and Tables

**Figure 1 healthcare-12-00224-f001:**
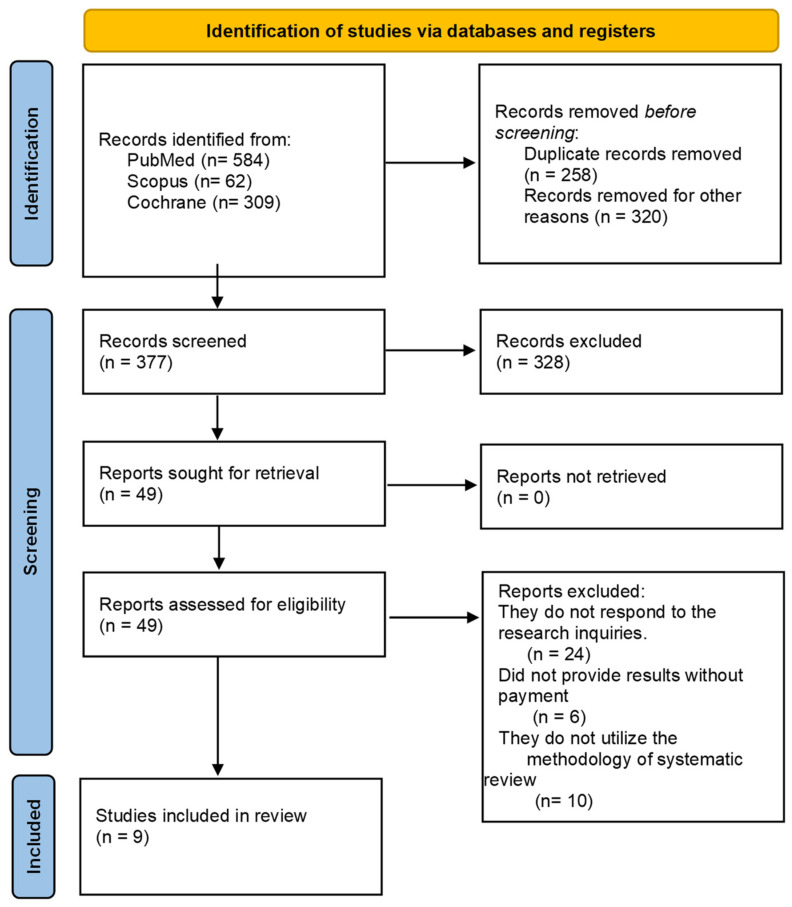
Flow diagram of the selection of publications for review.

**Figure 2 healthcare-12-00224-f002:**
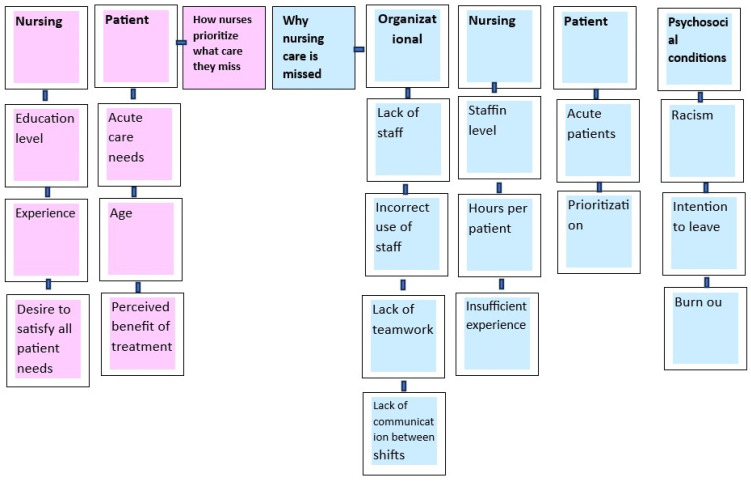
Factors influencing why nurses miss care and the criteria guiding their decisions on which care to prioritize.

**Table 1 healthcare-12-00224-t001:** Eligibility Criteria.

	Eligibility Criteria	
	Inclusion	Exclusion
Research inquiries	Why nursing care is missed? How nurses prioritized what care they miss?	Documents that did not answer the research inquiries.
Context	All nurses/patients are included regardless of educational level and workplace.	Documents that included and other health care workers (doctors, dentists etc.) and had non-obvious results
Article type	Systematic reviews	All other kind of reviews
Text availability	Free full text	Paid content
Language of publication	English	Documents that were not available in English
Publication date	2013–2023	Before 2013

**Table 2 healthcare-12-00224-t002:** Indexed terms selection strategy.

Database	Indexed Terms	Records Retrieved
Scopus	Missed nursing care	26
	Unmet nursing care	10
	Unfinished care	6
	Rationed nursing care	3
	Prioritized nursing care	13
	Omissions in nursing care	4
	Nursing care left undone	0
PubMed	Missed nursing care	175
	Unmet nursing care	147
	Unfinished care	14
	Rationed nursing care	82
	Prioritized nursing care	128
	Omissions in nursing care	31
	Nursing care left undone	7
Cochrane	Missed nursing care	294
	Unmet nursing care	2
	Unfinished care	0
	Rationed nursing care	3
	Prioritized nursing care	6
	Omissions in nursing care	4
	Nursing care left undone	0
Total		955

**Table 3 healthcare-12-00224-t003:** Characteristics of included studies (n = 9).

Author (Year)	Country	Review Type	Databases Used	Number of Studies	Focus/Aim	Quality Appraisal	Sample Size
Griffiths P, Recio-Saucedo A, Dall’Ora C (2018) [31]	UK	Systematic review of quantitative studies	CEA registry, CDSR, CENTRAL, CINAHL, DARE, Econlit, Embase, HTA database, Medline including In-Process, NHS EED, HEED and databases of grey literature (including the HMIC database and those held by the National Institute for Health and Care Excellence [NICE])	N = 18 (database search n = 11,269)	1. What are the nursing care tasks most frequently missed in acute hospitals adult inpatient wards, as reported by staff or patients, or captured in administrative data? 2. What are the associations between missed care and nurse staffing levels or skill mix in acute hospitals’ adult inpatient wards?	Adapted the NICE quality appraisal checklist for quantitative studies reporting correlations and associations (National Institute for Clinical Excellence	232–31,627 (RNs, HCSW)
Andersson I, Bååth C, Nilsson J, Eklund AJ. A scoping review-Missed nursing (2022) [32]	Sweden	Scoping review	CINAHL, PubMed and Scopus	N = 16 (database search n = 2714)	1. What characterized the studies in the area? 2. How was missed nursing care measured? 3. What was the content of the identified instruments and questions? 4. Are the identified instruments validated, and if so, how? 5. What were the main findings of the studies?	Quality appraisal was conducted on the papers according to the Guide to an Overall Critique of a Quantitative/Qualitative Research Report	264–4847
Imam A, Obiesie S, Gathara D (2023) [33]	UK	Systematic review	Medline, Embase, Global Health, WHO Global index medicus and Cumulative Index to Nursing and Allied Health Literature (CINAHL)	N = 7 (database search n = 1248)	1. To determine the prevalence of missed nursing care and the categories of nursing care that are most frequently missed in acute hospital settings in LMIC. 2. To document the factors associated with and reasons for missed nursing care in LMIC settings.	Used the Newcastle–Ottawa Scale	28–7802 (nurses or midwives. Excluded studies that examined missed care among other cadres of healthcare professionals including nurse assistants)
Duhalde H, Bjuresäter K, Karlsson I (2023) [34]	Sweden	Scooping review	Ovid Medline, Cumulative Index to Nursing and Allied Health Literature (Cinahl), and Web of Science (WOS).	N = 55 (database search n = 3557)	What is the prevalence, and causes of MNC in the context of EDs? What is known about MNC in relation to patient safety and quality of care in the context of EDs?	Not reported	9141 (only RN’s)
Chiappinotto S, Papastavrou E, Efstathiou G (2022) [35]	Greece	Systematic review	MEDLINE, the Cumulative Index to Nursing and Allied Health Literature (CINAHL), and SCOPUS	N = 58 (database search n = 1120)	The aims of the study were to (a) map factors, predictors, correlates, or linked factors—hereafter, ‘antecedents’, and (b) summarize the direction of their relationships with UNC.	Te 58 studies were evaluated for their methodological quality with the Joanna Briggs Institute Critical Appraisal approach	Not reported
Hilario C, Louie-Poon S, Taylor M (2023) [36]	Canada	Scoping review	Medline, EMBASE, PsycINFO, CINAHL, Alternative Press Index, Anthropology Plus, International Journal of Social Determinants of Health and Health Services, Scopus Elsevier	N = 13 (database search n = 4000)	1. What are the characteristics of the literature examining racism in health service use for adolescents? 2. What are the foci of the literature on racism and health services for adolescents?	Not reported	Not reported
Gustafsson N, Leino-Kilpi H, Prga I (2020) [37]	Finland	Scoping review	PubMed, CINAHL, PsycINFO, Web of Science, ProQuest and Philosophers Index	N = 13 (database search n = 2145)	1. How have patients’ perceptions of missed care been studied? 2. What instruments were used to measure patients’ perceptions on missed care? 3. What were the main findings of the studies? 4. What are the implications and suggestions for further research in the studies?	Not reported	352–66,348 patients
Stemmer, R., Bassi, E., Ezra (2021) [38]	Germany	Systematic review	CINAHL, the Cochrane Library, Embase, Medline, ProQuest and Scopus	N = 9 (database search n = 228)	To investigate the association of unfinished nursing care on nurse outcomes	Mixed Methods Appraisal Tool	136–4169 nurses
Suhonen, R., Stolt, M., Habermann, M. (2018) [39]	Finland	A scoping review	CINAHL and MEDLINE.	N = 25 (database search n = 2024)	To explore and illustrate the key aspects of the ethical elements of the prioritization of nursing care and its consequences for nurses.	Not reported	Not reported

**Table 4 healthcare-12-00224-t004:** Outcomes of why nursing care is missed and how nurses prioritize what care they miss.

Author (Year)	Why Nursing Care Is Missed
Griffiths P, Recio-Saucedo A, Dall’Ora C (2018) [31]	Staffing level/Less nursing hours per patient
Andersson I, Bååth C, Nilsson J, Eklund AJ. A scoping review-Missed nursing (2022) [32]	Lack of staff/Incorrect use of staff/Prioritization
Imam A, Obiesie S, Gathara D (2023) [33]	Inadequate number of nursing staff/Inadequate number of assistive personnel and unexpected rise in patient volume and/or acuity both ranked
Duhalde H, Bjuresäter K, Karlsson I (2023) [34]	Having to take care acute patients/not time for basic care needs/Prioritization
Chiappinotto S, Papastavrou E, Efstathiou G (2022) [35]	Not reported
Hilario C, Louie-Poon S, Taylor M (2023) [36]	Racism
Gustafsson N, Leino-Kilpi H, Prga I (2020) [37]	Insufficient experience/Lack of teamwork/Lack of communication between shifts/Prioritization
Stemmer, R., Bassi, E., Ezra (2021) [38]	Intention to leave the job/Burn out
Suhonen, R., Stolt, M., Habermann, M. (2018) [39]	Prioritization

**Table 5 healthcare-12-00224-t005:** Outcomes of and how nurses prioritize what care they miss.

Author (Year)	How Nurses Prioritize What Care They Miss
Griffiths P, Recio-Saucedo A, Dall’Ora C (2018) [31]	Not reported
Andersson I, Bååth C, Nilsson J, Eklund AJ. A scoping review-Missed nursing (2022) [32]	Not reported
Imam A, Obiesie S, Gathara D (2023) [33]	Not reported
Duhalde H, Bjuresäter K, Karlsson I (2023) [34]	Based on their education and experience/Prioritize patients acute care needs
Chiappinotto S, Papastavrou E, Efstathiou G (2022) [35]	They desire to provide the best care for their patients and eliminate unfinished care
Hilario C, Louie-Poon S, Taylor M (2023) [36]	Not reported
Gustafsson N, Leino-Kilpi H, Prga I (2020) [37]	Based on their education and experience
Stemmer, R., Bassi, E., Ezra (2021) [38]	Not reported
Suhonen, R., Stolt, M., Habermann, M. (2018) [39]	Based on a desire to satisfy all the needs of their patients in a holistic and comprehensive manner/Appeared in a number of contexts: patients group, specific disease, severity of patients situation, age, perceived benefit of treatment

## Data Availability

Data are contained within the article and Appendix A.

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
