# Peer review of "Missed Nursing Care; Prioritizing the Patient’s Needs: An Umbrella Review"

_healthcare, 2024, doi:10.3390/healthcare12020224_

Round 1

Reviewer 1 Report

Comments and Suggestions for Authors

The article in question addresses a relevant issue and proposes the development of a comprehensive but systematic analysis. The objectives are clear and the description of the methodological options and procedures are duly supported. Overall, it makes a relevant contribution to understanding the phenomenon under study, although there are also some weaknesses.

The following are the main critical aspects:

Part of the introduction - namely between lines 53 and 81 - is presented in an excessively schematic way. A more elaborate textual development would be important, so that the text is not just a succession of topics that are poorly articulated.

 The same could be said of the beginning of section 2 - between lines 108 and 111 - as point 2.1 is presented in an excessively summarised and schematic way.

With regard to point 2.2, it is important to clarify the reasons behind some of the criteria presented. In particular, why the time period 2013-2023? why are paid content not taken into account? and why only articles in English?

As far as table 3 is concerned, it should be noted that it lacks the necessary legibility and should therefore be improved.

Finally, it should be noted that the article's conclusion is too brief and does not fulfil any differentiating role. The little that is said doesn't add anything relevant, which is why further critical elaboration is recommended in this section. It would clearly be appropriate to reflect on the limits and potential of this work, and it would also be interesting to give some concrete examples that try to show how some interventions based on scientific literature and empirical evidence have, or have not, led to improvements in the reality analysed here.

Author Response

Thank you for your comments. Please find our answer.

Reviewer 2 Report

Comments and Suggestions for Authors

Dear All,

It was with great interest that I read the paper entitled ‘Missed nursing care; Prioritizing the patient's needs: An Umbrella Review’. The work constitutes an extremely valuable voice in the scientific discussion that allows to prove how extremely important is  the right choice of nursing staff for the entire health care system.

Having analysed the paper, I reached the following conclusions:

1.               The content of the work is consistent with the aim of the work.

2.               The Introduction is exhaustive, however, the aim of the work should be mentioned at the end of the section.

3.               The methodology does not raise any doubts.

4.               The Results section was written in a comprehensible way.

5.               The Discussion does not include references to other papers. In its current form, it resembles more closely summary, rather than discussion.

6.               There is no need to refer to the objectives of the paper in the Conclusion section.

7.               The research literature was correctly selected.

Author Response

(The authors gave the same response as above.)

Reviewer 3 Report

Comments and Suggestions for Authors

This article delves into a crucial and pressing issue that demands exploration, and the overall quality of the manuscript is commendable. However, there are critical aspects that currently hinder a comprehensive assessment of the study. The tables provided, unfortunately, are in a state that impedes their assessment, rendering them difficult to interpret or analyze effectively. To comprehensively evaluate the findings, supplementary materials that should have been included with the submission are notably absent. This absence significantly limits the ability to gauge the depth and accuracy of the research.

For a thorough and rigorous evaluation of the study, it is highly recommended that the authors resubmit the manuscript inclusive of all the supplementary materials required. By providing these additional materials, such as the missing tables or other essential documents, the scholarly community will have the necessary resources to properly assess the study's validity, replicability, and overall contribution to the field. This resubmission will not only facilitate a more comprehensive evaluation but also enhance the study's impact and credibility within the academic sphere.

Author Response

(The authors gave the same response as above.)

Reviewer 4 Report

Comments and Suggestions for Authors

First congratulate the researchers for the peculiar article, and at the same time for the opportunity to review them, it is an interesting article since the lack of nursing care is associated with patient safety and the quality of nursing care provided.

these are some contributions to the improvement of the article.

The introduction, in this section we clearly indicate the justification of the research (why it is done), summarize the criteria that have led to its realization and provide the minimum bibliographical substratum essential to expose the updated state of other researches in relation to the same object of study and why it is necessary to continue researching and publishing on it. 

Clearly describe the selection of studies, detailing eligibility and exclusion criteria and a description.

Table 3, 4  and 5, is not clear, I do not know if it is due to the downloading of the article for the review, or it has not been presented correctly but the data are not seen, please put these tables better, for a clarifying reading of the study.

In the last paragraph of the introduction I would have liked to find the objective of the research, which I recommend adding.

the conclusions are very few, I think that we could go a little deeper. 

I do not find the protocol if it has passed an ethics committee, for a future study after the review, or it will only remain a review.

please review and update the bibliography. 

please review and update the bibliography. 

Author Response

(The authors gave the same response as above.)

Reviewer 5 Report

Comments and Suggestions for Authors

Thank you for having the chance to review this interesting study. I think that this manuscript cannot be published without revision. Please check out the following:

If registered, provide the name of the registry (such as PROSPERO) and registration number.

Introduction:

Line 66, “In the context of MNC, the nursing process is described as a client-centered approach used to organize nursing care.” I am not sure what this sentence means.

I think the contents in the introduction are not distinguished from those in the discussion.

In addition, you had better describe the rationale for the review in the context of what is already known.

Results:

Page 8, The contents of “why nursing care is missed” are the same as those of “why care is missed”. I am not sure if both are necessary.

Conclusions:

Provide a general interpretation of the results with respect to the review questions and objectives, as well as potential implications.

Comments on the Quality of English Language

 Minor editing of English language required.

Author Response

(The authors gave the same response as above.)

Round 2

Reviewer 3 Report

Comments and Suggestions for Authors

The authors didn't take into consideration the comments given in the previous review.

Author Response

Thank you for your comments. We provide supplementary material to enhance the study's impact and credibility by creating a table with a quality appraisal for included reviews using the Joanna Briggs Institute (JBI) tool. Please see the attachment.

Reviewer 5 Report

Comments and Suggestions for Authors

Thank you for having the chance to review this interesting study. I think that this manuscript was revised according to review comments. 

Author Response

Thank you for your comments. We addressed all issues from the first round